# Novel Intronic Mutation in *VMA21* Causing Severe Phenotype of X-Linked Myopathy with Excessive Autophagy—Case Report

**DOI:** 10.3390/genes13122245

**Published:** 2022-11-29

**Authors:** Antoine Pegat, Nathalie Streichenberger, Nicolas Lacoste, Marc Hermier, Rita Menassa, Laurent Coudert, Julian Theuriet, Roseline Froissart, Sophie Terrone, Francoise Bouhour, Laurence Michel-Calemard, Laurent Schaeffer, Arnaud Jacquier

**Affiliations:** 1Service ENMG et Pathologies Neuromusculaires, Hôpital Neurologique P. Wertheimer, Hospices Civils de Lyon, 69500 Bron, France; 2Pathophysiology and Genetics of Neuron and Muscle, CNRS UMR 5261, INSERM U1315, INMG, Université Claude Bernard Lyon 1, Faculté de Médecine Lyon Est, 69008 Lyon, France; 3Service d’anatomopathologie, Centre de Biologie et Pathologie Est (CBPE), Hospices Civils de Lyon, 69500 Bron, France; 4Service de Neuroradiologie, Hôpital Neurologique P. Wertheimer, Hospices Civils de Lyon, 69500 Bron, France; 5Service de Biochimie et Biologie Moléculaire, Centre de Biologie et Pathologie Est (CBPE), Hospices Civils de Lyon, 69500 Bron, France; 6Centre de Biotechnologie Cellulaire, CBC Biotec, Hospices Civils de Lyon-Groupement Est, 69500 Bron, France

**Keywords:** XMEA, VMA21, intronic mutation, intron retention, autophagy, vacuolar myopathy

## Abstract

X-linked Myopathy with Excessive Autophagy (XMEA) is a rare autophagic vacuolar myopathy caused by mutations in the Vacuolar ATPase assembly factor *VMA21* gene; onset usually occurs during childhood and rarely occurs during adulthood. We described a 22-year-old patient with XMEA, whose onset was declared at 11 through gait disorder. He had severe four-limb proximal weakness and amyotrophy, and his proximal muscle MRC score was between 2 and 3/5 in four limbs; creatine kinase levels were elevated (1385 IU/L), and electroneuromyography and muscle MRI were suggestive of myopathy. Muscle biopsy showed abnormalities typical of autophagic vacuolar myopathy. We detected a hemizygous, unreported, intronic, single-nucleotide substitution c.164-20T>A (NM_001017980.4) in intron 2 of the *VMA21* gene. Fibroblasts derived from this patient displayed a reduced level of *VMA21* transcripts (at 40% of normal) and protein, suggesting a pathogenicity related to an alteration of the splicing efficiency associated with an intron retention. This patient with XMEA displayed a severe phenotype (rapid weakness of upper and lower limbs) due to a new intronic variant of *VMA21*, related to an alteration in the splicing efficiency associated with intron retention, suggesting that phenotype severity is closely related to the residual expression of the VMA21 protein.

## 1. Introduction

Autophagic vacuolar myopathies are characterized by the presence of autophagic vacuoles observable on muscle biopsy samples, and are either due to genetic factors (Pompe Disease (Glycogen Storage Disease type II), Danon Disease, and X-linked Myopathy with Excessive Autophagy (XMEA)) or related to acquired diseases (especially inclusion body myositis and chloroquine toxicity) [1,2].The onset of XMEA usually occurs during childhood and rarely occurs during adulthood [2]. XMEA is characterized by slowly progressive proximal weakness with the preservation of ambulation, with no extramuscular or cardiac abnormality [2]. This myopathy was first described in 1988 [3], and the genetic cause, mutations in the Vacuolar ATPase assembly factor *VMA21* gene, was identified in 2013 [4]. Serum creatine kinase levels are frequently very elevated, sometimes >4000 IU/L [2]. Muscle biopsy histopathology is characterized by an abnormal number of cytoplasmic vacuoles, by the deposition of complement C5b-9 within vacuoles and along the sarcolemma, and by the expression of MHC class I [2]. This myopathy only affects males as it is due to mutations in the *VMA21* gene located at Xq28 [4]. *VMA21* mutations impair vacuolar-ATPase assembly, thereby leading to a decreased enzyme activity, a reduction in lysosomal acidification, and a blockage of autophagy [4].

VMA21 mutations have mainly been identified in introns. Some affect the splicing branch point (c.54-27A>C; c.54-27A>T; c.54-16_54-8del) [4], some affect the splicing donor site (c.163 + 4A>G) [4,5], and some affect the splicing acceptor site (c.164-6T>G; c.164-7T>G) [4,6,7,8]. Only one mutation was identified in an exon (c.272G>C), but this point mutation was predicted to affect a splice-enhancer site [4,9]. Finally, two mutations were identified at the beginning of the 3’UTR (c.*6A>G; c.*13_*104del), but their mechanism of action remains unknown. However, all of these mutations lead to a reduction in *VMA21* mRNA and VMA21 protein levels [2,4,9]. We described herein a patient with XMEA due to a new intronic causative variant who presented a severe phenotype.

## 2. Case Report

### 2.1. Clinical and Paraclinical Summary

A 16-year-old man presented pure proximal four-limb weakness, for which the onset occurred at the age of 11 with a gait disorder and tip-toe walking. At 16, he had a waddling walk with tip-toe and genu recurvatum, over less than 50 m. Clinical examination found severe four-limb proximal weakness and amyotrophy, prominent in the lower limbs. Muscle strength using the Medical Research Council (MRC) scale was 4+/5 on arm abduction and elbow flexion, and 2/5 on hip flexion and abduction and knee extension. He had Achilles tendon contractures. He had bilateral symmetrical scapular winging and axial weakness. He had no sensory symptoms/signs and normal deep tendon reflexes except for abolished rotulian reflexes. No abnormality was noticed on the cephalic extremity. Cognitive development was normal. He had no cardiac abnormality. He had low vital capacity at 69% (respiratory functional exploration), without dyspnea. The serum creatine kinase level was elevated: 1385 IU/L (normal <200 IU/L). Electromyography showed abundant spontaneous activity (fibrillations and complex repetitive, myotonic discharges at rest) and clear myogenic changes during effort. Ultrasonography showed homogeneous muscular hyperechogenicity predominantly in the quadriceps (Figure 1A). Magnetic resonance imaging (MRI) of the pelvis and upper thighs and legs showed bilateral symmetric muscular fatty replacement without muscle edema or contrast enhancement (Figure 1B,C). Four years later, at age 22, upper limb weakness increased (deltoids, biceps MRC scale 3/5; grip strength (using handheld dynamometry) about 40% of the normal level (around 20 kg)), and walking was possible over about 50 m with a knee orthesis for a genu recurvatum, but most of the time, he used an electric wheelchair.

### 2.2. Muscle Biopsy

Muscle biopsy (left deltoid) was performed at the age of 16 years. After isopentane freezing the sections, histological and histoenzymatic techniques showed non-grouping, atrophic, round fibers and multiple nucleus internalizations; interestingly, numerous cytoplasmic autophagic vacuoles were observed and were strongly positive for PAS and phosphatasic acid. Immunohistochemistry stainings (p62 and TDP 43) were positive in cytoplasmic autophagic vacuoles. Vacuolated fibers expressed MHC class I and complement C5b-9 without any inflammation infiltrates (Figure 1D–G). In conclusion, a diagnostic of autophagic vacuolar myopathy was proposed.

### 2.3. Genetic Analyses

As an autophagic vacuolar myopathy was suspected, maltase acid activity (Pompe Disease) was assessed and exon regions and junctions (+/−25 nt) of the *LAMP2* and *GAA* genes (Danon Disease and Pompe disease respectively) were sequenced, and both were normal.

Peripheral blood was sampled from the patient after informed consent. A molecular analysis of the *VMA21* gene was performed by the amplification of exons and flanking intronic regions using PCR followed by bi-directional capillary sequencing on ABI3500XL (AppliedBiosystems). The sequences produced were compared to the human reference genome (GRCh37/hg19) with the SeqScape v3 software (Thermo Fisher Scientific Inc.). Primer sequences are available upon request.

Sanger sequencing of the VMA21 gene detected hemizygous, unreported, intronic, single-nucleotide substitution c.164-20T>A (NM_001017980.4) in the last intron. This variant has never been described to our knowledge (absent from gnomAD and HGMD). Its in silico analysis predicted a loss of strength of the acceptor site located 20 bp downstream. Indeed, Alamut Visual Plus (v1.6.1 from Sophia Genetics) predicted a negative impact based on four algorithm predictions (GeneSplicer: −25%; MaxEntScan: −18%; SpliceSiteFinder-like: 0%; NNSPLICE: 0%). Additionally, Splice AI predicted an acceptor loss with a score of 0.42 in a scale from 0 to 1 which can be interpreted as the probability that the variant affects the splicing [10].

The patient’s maternal uncle displayed no symptoms or clinical signs and did not carry this mutation. The asymptomatic patient’s mother was not tested.

This variant was classified as likely pathogenic according to the American College of Medical Genetics (ACMG) guidelines (PS3, PM2, PP3; in vitro functional studies, absence from databases, in silico analysis predicting a splicing impact) [11].

### 2.4. mRNA and Protein Quantification from the Patient’s Fibroblasts

The *VMA21* gene (Ensembl:ENSG00000160131; MIM:300913; AllianceGenome:HGNC:22082) encodes three different transcripts. Only two of them (VMA21–201; 4715nt, ENST00000330374.7, NM_001017980.4 and VMA21–202; 693nt, ENST00000370361.5) are predicted to be translated into 101- and 156-amino-acid-long proteins, respectively (Figure 2A). To investigate the impact of the mutation c.164-20T>A, we used fibroblasts from the case-report patient (AC969 line) and two wild-type control fibroblasts (AB249 and V972 lines) derived from a skin biopsy. Indeed, VMA21 expression is enriched in fibroblasts, and the VMA21-201 transcript is the most abundant in this cell type (gtexportal.org accessed on 1 November 2022). To determine the abundance of *VMA21* mRNA, three primer pairs were designed (Appendix A). Two of them amplified specifically the two translated transcripts VMA21–201 and VMA21–202, and the third pair of primers was designed to amplify the three transcripts indifferently by flanking the intron that carries the mutation. Quantitative RT-PCR found using these three pairs of primers, a significant 60% reduction in *VMA21* transcript levels in AC969 fibroblasts (patient) was seen compared to the two wild-type control fibroblasts (*n* = 3; two-way ANOVA test; *** *p* < 0.0001; Figure 2B). To investigate the reduction in the acceptor splice site score predicted in silico, we designed a pair of primers that specifically amplified the junction between the mutated intron and the previous exon that are common to all transcripts (Figure 2A). Quantitative RT-PCR found a significant 300% increase in *VMA21* intron retention transcript levels in AC969 fibroblasts (patient) compared to the two wild-type control fibroblasts, suggesting a default in the maturation that could trigger mRNA decay (*n* = 3; one-way ANOVA test; *** *p* < 0.0001; Figure 2C). Altogether, these results suggest that the mutation impacts the correct splicing of the mutated intron, leading to reduced stability of *VMA21* mRNA.

To determine whether VMA21 protein levels were also reduced in the patient, a Western Blot was performed on protein extract from the AC969 fibroblasts and from a wild-type control fibroblast line (V972 line; Appendix A). The amount of α-tubulin and histone H4 were used as loading controls (Figure 2D). Anti VMA21 antibody revealed a significant reduction in VMA21 protein levels in the AC969 fibroblasts (patient), suggesting that the variant is pathogenic.

## 3. Discussion

We described herein a new intronic causative variant in the *VMA21* gene harbored by a patient with XMEA. Both the mRNA and protein levels were greatly reduced in this patient, suggesting the pathogenicity of this mutation, related to an alteration in the splicing efficiency associated with intron retention. The patient displayed classical specificities of XMEA such as the absence of extramuscular or cardiac sign and a typical muscle biopsy of vacuolar myopathy with excessive autophagy, with a highly elevated serum creatine kinase level (about 7× the normal level) [2]. He had myotonic discharges, as described especially in dystrophy myotonic type 1 or 2, Pompe disease, or acquired myopathies [12]. Myotonic discharges have already been described in the first XMEA patients [3] and later confirmed [13,14]. Muscle MRI also showed a fatty infiltration, as previously described [15], and confirmed here via ultrasonography. 

However, although most XMEA patients display a very slow progression of weakness and are still ambulant over the age of 50 [2], the patient presented here displayed a rapid progression of weakness with great involvement of the upper limbs in 4 years and significant difficulties walking despite his young age. An allelic disease of XMEA, named congenital autophagic vacuolar myopathy (CAVM), has also been described in a few patients and has been associated with a severe phenotype (death or need of intubation–ventilation and nasogastric feeding) [6]. This disease is caused by a specific intronic mutation (c.164-6T>G) in *VMA21* which reduces the splicing efficiency, thereby reducing *VMA21* mRNA expression to 22–25% of normal expression [6]. Classical XMEA patients display *VMA21* mRNA levels which are 42–69% of normal expression [4,6,9]. The patient presented herein had a more severe phenotype compared to classical forms of XMEA, which could be related to the intronic variant he harbored and led to a great reduction in *VMA21* mRNA level at 40% of normal expression. The severity of diseases (XMEA/CAVM) due to the *VMA21* variant seems to be mainly related to the residual level of *VMA21* mRNA correctly spliced, which seems itself to be related to the mutation in *VMA21*.

## 4. Conclusions

To conclude, we described a patient with XMEA displaying a severe phenotype (rapid weakness of four limbs) due to a new intronic variant of *VMA21*, suggesting that the phenotype severity is closely related to the residual expression of VMA21 protein.

## Figures and Tables

**Figure 1 genes-13-02245-f001:**
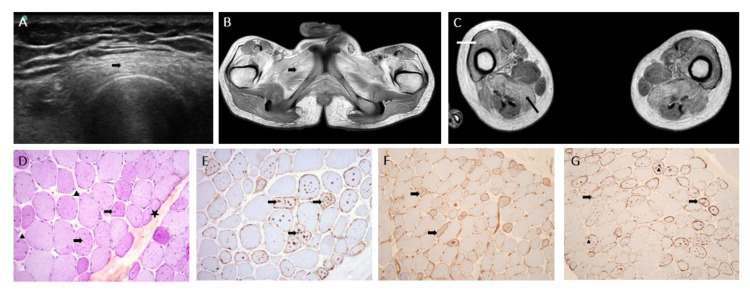
Imaging and muscle biopsy. (**A**) Muscle ultrasonography of the left thigh (at age 22 years) showing hyperechogenicity of quadriceps muscles (arrow). (**B**) T1-weighted magnetic resonance imaging (MRI), transversal section of the pelvis showing moderate amyotrophy, and hypersignal of muscle glutei and adductors (arrow) consistent with fatty infiltration. (**C**) T1-weighted MRI, transversal section of the thigh showing fatty infiltration of muscles predominant in anterior (white arrow; vastus lateralis, vastus medialis, vastus intermedius, rectus femoris) and posterior (black arrow; hamstring muscles) compartments. (**D**) PAS ×20: star: atrophic fibers, arrow head: fibrosis, arrow: numerous glycogenic vacuoles. (**E**) p62 ×20: arrow: autophagic vacuoles. (**F**) MHC class I immunostaining: arrow: membranous expression of vacuolated fibers. (**G**) C5-b9 immunostaining: arrow head: intravacuolar deposits, arrow: membranous expression.

**Figure 2 genes-13-02245-f002:**
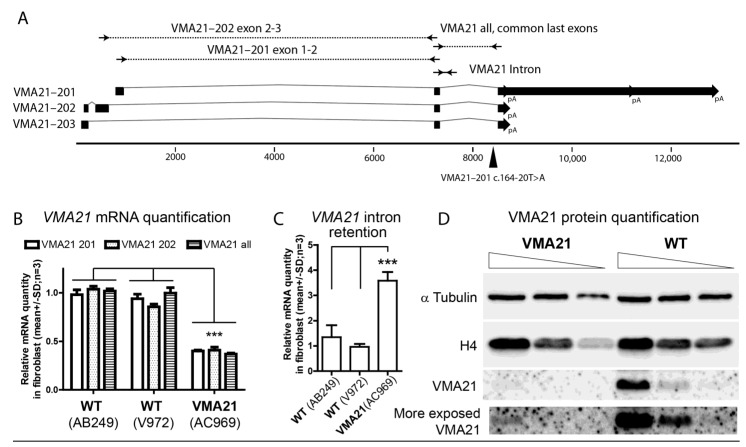
VMA21 causative variant and mRNA- and protein-level quantification. (**A**). Schematic representation of *VMA21* transcripts (Ensembl database). Squares represent exons, lines represent introns, and block arrows represent the polyadenylation site and it alternative sites (pA). Primers are represented by arrows. Primer pairs VMA21–201 exon1-2 and VMA21–202 exon1-2 specifically amplify each corresponding isoform, whereas the primer pair *VMA2*1 exon2-3 amplifies all three isoforms. The mutation c.164-20T>A affecting isoform 1 (VMA21–201 or NM_001017980.4 or ENST00000330374.7) located in the last intron is represented by a black triangle. (**B**). *VMA21* mRNA quantification via qRT-PCR on wild-type control fibroblasts (line AB249 and line V972) compared to fibroblasts derived from the case-report patient (line AC969). The primer pairs illustrated in (**A**) were used. For all three primer pairs used, the levels of *VMA21* mRNA were significantly lower in line AC969 compared to the wild-type control cell lines. Two-way ANOVA test. *** *p* < 0.0001 (**C**). *VMA21* intron retention quantification via RT-PCR on wild-type control fibroblasts (line AB249 and line V972) compared to fibroblasts derived from the case-report patient (line AC969). The levels of intron retention in *VMA21* mRNA were significantly higher in the cell line AC969 compared to the wild-type control cell lines. One-way ANOVA test. *** *p* < 0.001 (**D**). VMA21 protein quantification via Western Blot on AC969 fibroblasts versus wild-type control fibroblasts (line V971). Three concentrations (4×, 2×, and 1×) of protein extract were loaded for each line. α-tubulin and histone H4 were revealed and used as loading control. VMA21 antibody detected a clear signal in the wild-type cell line.

## Data Availability

The data presented in this study are available on request from the corresponding author.

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
