# Peer review of "Novel Intronic Mutation in VMA21 Causing Severe Phenotype of X-Linked Myopathy with Excessive Autophagy—Case Report"

_genes, 2022, doi:10.3390/genes13122245_

Round 1

Reviewer 1 Report

The article by Pegat et al “ Novel deep intronic …………………case report” is well written and defined the clinical summary of the case (MRI, biopsy, biochemical) and the experimental findings (genetic analysis , mRNA and protein quantification) in very concise manner to prove that one of the intronic variant observed in this patient  is pathogenic and presenting a severe phenotype. Results presented in this article suggest that intronic variant can impact the mRNA and protein of VMA21 gene, which could have functional consequences. Additionally, autophagic vacuole and MHC1 data presented by IHC on the patient biopsy sample.

Here are my comments:

Since high creatine and autophagy is very commonly associated with  lysosomal storage disorders along with the myopathy, authors should check some of the autophagy markers through real time PCR or Western blot to rule out the possibility of additional disorders in this patient. Late onset lysosomal storage disorders patient sometime has similar clinical feature and symptoms as late onset myopathy like XMEA.

Authors should indicate which Insilco tool/s  they have utilized to study  the impact of this variant on the splicing ( reduction in acceptor site).

Since this variant is within 25 base pair from exon-intron junction, I will suggest authors to remove terminology “Deep intronic variant” for this variant as deep intronic variants are those variants which mostly affect the transcriptional machinery (enhancer etc), and mostly they impact non-canonical splice sites. Authors should use the term just “intronic variant”.

How the authors described this variant as a pathogenic is not clear. Most of the variant should classify based on the ACMG criteria to make a call for pathogenic or other criteria. Authors should explain what criteria they used to call this variant as pathogenic despite all the functional studies and clinical feature suggest this variant could be causative source for XMEA in this particular patient. If it is not clear, authors should rephrase some of the sentences in the article from pathogenic variant to deleterious or causative variants.

Reviewer 2 Report

This manuscript claims that a "deep-intronic" mutation in the last intron of the VMA21 gene is responsible for a rare X-linked myopathy with excessive autophagy.

This is a case report and hence it is important that the mutation is clearly identified and characterized.

The manuscript needs to supply many more details in background,  techniques,  procedures and analysis before I would be convinced that this c.164-20T>A change is truly pathogenic. 

A minor point.., I do not regard 20 bases upstream of a 3' splice (acceptor) site as "deep".  deep intronic mutations are more like hundreds or thousands of bases away from the exons.

From the data provided, there is clearly a loss of the VMA21 protein in the proband, so a catastrophic mutation appears to have occurred.  

Line 47.  Could the authors expand on the types of mutations found to date... Missense or frame-shifting indels that disrupt the reading frame or nonsense mutations.

Line 105 ... sentence needs correction but I doubt that the entire 43,000 bp of the LAMP2 gene was sequenced.

Line 115.  What in silico program was used to calculate the score.  Some programs I have seen would not even include any changes 20 bases from the acceptor site.  Does the -9% refer to loss of strength?

Line 119. Please explain why the mother was not tested.  If she did not carry the mutation then it could have arisen de novo.

I would regard showing the RT-PCR gel amplicons/results as essential.  This is a relatively small transcript and when the authors claim a 40% reduction in mRNA levels (compared to normal cell lines) Fig 2B , I was wondering why there was not a 40% reduction in protein (rather than absolute loss on the western Fig 2D).

It would help to include primer sequences and the Fig1A (all common last exons) appears to show the reverse primer very close to the beginning of the last exon/end of the intron.  I assume the block arrows indicate the polyadenylation sites.. perhaps this should be mentioned

Can the authors explain why there is any detectable intron in the normal cells in Fig C.  Could this be from DNA contamination.  There was no mention of PCR conditions, cycle number, qRT-PCR conditions etc and RNA extraction.

There is approximately a 3 fold increase in intron retention in the patient cells. If the intron was retained, I would have expected much a much more pronounced signal from the patient.

Why was RT-PCR not undertaken at the end of that intron so that the mutation was included?

Did the authors undertake long range RT-PCR from some of the early exons across to the last exon.  The intron is only some 1175 bases long, well within the range of XL-PCR systems.

Did the authors sequence the patient RT-PCR amplicons generated to confirm no subtle changes.  The T>A change occurs just before a G and could possibly generate a weak cryptic splice site missing 20 bases (hence the request to look at the RT-PCR bands, with appropriate controls (ie no RT step or no template). This size difference should have been seen in the all common last exon amps.

I cannot explain what has happened to the splicing of this VMA21 transcript from the data provided.  Low levels (~40%) of the normal transcript are being detected  by all amplifications, so where is the protein?  Did highly sensitive RT-PCR conditions have to be employed?  Addiiotnal details are needed.

More detail in experimental conditions, as well as some more RT-PCRs across the entire transcript are needed.  Was the entire 3'UTR sequences in the VMA21 201 transcript?

Round 2

Reviewer 2 Report

 A few minor spelling errors but my concerns are generally addressed.  It would be good if the authors did find the time to scan the rest of the transcript (from the first/last exon) using long range PCR to confirm there are no deep intronic mutations outside the area they have scanned.  If nothing is found, then they can be much more confident they have identified the causative mutation.

If they do find something, then another publication highlighting the need for thorough scanning of mRNAs.